# Emulsifier and Xylanase Can Modulate the Gut Microbiota Activity of Broiler Chickens

**DOI:** 10.3390/ani10122197

**Published:** 2020-11-24

**Authors:** Marta Kubiś, Paweł Kołodziejski, Ewa Pruszyńska-Oszmałek, Maciej Sassek, Paweł Konieczka, Paweł Górka, Jadwiga Flaga, Dorota Katarzyńska-Banasik, Marcin Hejdysz, Zuzanna Wiśniewska, Sebastian A. Kaczmarek

**Affiliations:** 1Department of Animal Nutrition, Poznan University of Life Sciences, Wołynska 33, 60-637 Poznan, Poland; marta.kubis@up.poznan.pl (M.K.); marcin.hejdysz@up.poznan.pl (M.H.); zuzanna.wisniewska@up.poznan.pl (Z.W.); 2Department of Animal Physiology, Biochemistry and Biostrructure, Poznan University of Life Sciences, Wołynska 35, 60-637 Poznan, Poland; pawelbigi@o2.pl (P.K.); ewa.pruszynska@up.poznan.pl (E.P.-O.); maciej.sassek@up.poznan.pl (M.S.); 3The Kielanowski Institute of Animal Physiology and Nutrition, Polish Academy of Sciences, Instytucka 3, 05-110 Jablonna, Poland; pawel_konieczka@hotmail.com; 4Department of Animal Nutrition and Biotechnology, and Fisheries, University of Agriculture in Krakow, Al. Mickiewicza 24/28, 30-059 Krakow, Poland; p.gorka@ur.krakow.pl (P.G.); jadwiga.flaga@urk.edu.pl (J.F.); 5Department of Animal Physiology and Endocrinology, University of Agriculture in Krakow, Al. Mickiewicza 24/28, 30-059 Krakow, Poland; dorota.katarzynska@urk.edu.pl; 6Department of Animal Breeding and Animal Product Quality Assessment, University of Life Sciences, Słoneczna 1, 62-002 Złotniki, Poland

**Keywords:** sialic acid, villus, *Clostridium* spp., viscosity, *Bifidobacterium* spp., *Lactobacillus* spp., broiler chickens gut microbiota activity, gut morphology

## Abstract

**Simple Summary:**

Modern broiler nutrition, due to widely accepted goals of sustainable production, is concerned with the improvement of nutrient utilization. To achieve this, in our study, we used feed additives that improve the value of feed components. Due to the significant amounts of non-starch polysaccharides (which are not digested under broiler intestinal tract conditions and negatively affect broilers performance) in popular feed components like wheat, enzymes are needed. Furthermore, the use of emulsifiers to improve fat digestion is necessary, as young birds do not secrete sufficient enzyme and bile salts. Previous studies have shown that an additional increase in carbohydrate digestibility can be obtained by using an emulsifier containing xylanase in the feed. Presumably, the increase in carbohydrate digestibility occurred after adding the emulsifier. In our study, we aimed to investigate the effect of xylanase, emulsifier, and a combination of both in wheat diets with high level of tallow on the gastrointestinal tract microbiota activity of 480 one-day-old male ROSS 308 broiler chickens. The simultaneous usage of both additives in wheat-based diets with beef tallow reduces the ileum microbiota activity and enhances cecum microbiota activity. Presumably, the addition of both additives results in a cumulative effect on the gut microbiota activity.

**Abstract:**

In this study, we aimed to investigate the effect of xylanase (XYL), emulsifier (EMU), and a combination of both (XYL + EMU) in wheat diet with a high level of tallow on gastrointestinal tract microbiota activity, excretion of sialic acids, and selected gut segments morphology of 480 one-day-old male ROSS 308 broiler chickens. The activities of bacterial enzymes in the ileal digesta were lower in experimental groups compared to the control (CON) group. Enzyme activity in the cecum was significantly higher than in the ileum. The additives did not affect the excretion of sialic acid. The number of duodenum goblet cells on the villi decreased in all of the experimental groups (*p* < 0.05). The simultaneous use of XYL + EMU deepened the ileum crypts (*p* < 0.05). The total short-chain fatty acid (SCFA) concentration in the cecal digesta was higher in experimental groups. The abundance of *Bifidobacterium*, *Lactobacillus,* and *Escherichia coli* did not change among experimental groups. The relative abundance of *Clostridium* was significantly (*p* < 0.05) lower in groups with emulsifier addition. In conclusion, the simultaneous usage of EMU and XYL in wheat-based diets with beef tallow reduces the ileum microbiota activity and enhances cecum microbiota activity. Presumably, the addition of both additives results in a cumulative effect on the gut microbiota activity.

## 1. Introduction

Modern broiler breeding strains are characterized by rapid growth and a high feed conversion rate. To use this high genetic potential, appropriate, precisely adapted nutrition needs to be used. Chicken feeding costs can constitute up to 80% of the total production costs [1], so to maintain the profitability of production while maintaining product quality, improving the utilization of nutrients contained in the feed would increase profits. This can be achieved by using feed additives that improve the value of feed components.

Wheat is the main cereal used in poultry nutrition in Europe. It is well known that wheat contains significant amounts of non-starch polysaccharides (NSPs)—mainly arabinoxylans, which negatively affect the production results of broilers [2,3]. NSPs are not digested by birds and also increase the viscosity of the gut content, which limits the use of feed energy and nutrients [4]. This directly affects the deterioration of production results, adversely affects the symbiotic intestinal microbiome, intestinal physiology, and indirectly affects the immune system of birds [5,6]. Diets consisting mainly of wheat containing a high level of indigestible, soluble NSP have been shown to increase the intestinal bacterial activity associated with poor broiler performance [7] and promote the proliferation of *Clostridium perfringens*, which can cause necrotic enteritis in young birds [8]. To prevent this, xylanase (an enzyme that breakdowns long NSP chains) is added to feed that contains significant amounts of wheat [9]. Xylanase has a positive effect on the digestibility and absorption of nutrients by reducing the viscosity of the ileum contents [10,11]. The decrease in digesta viscosity also affects the reduction of microbiome activity in the upper parts of the gastrointestinal tract [12,13], and the degradation of long NSP chains provides a substrate for bacteria existing in its lower sections [14,15].

Another way to increase the profitability of poultry production is to improve the availability of the most expensive feed compound ingredient, i.e., fat by using emulsifiers [16]. The gastrointestinal tract of a chicken is an aqueous environment. Fats, such as hydrophobic components, must aggregate to form micelles to be absorbed. Emulsifiers found in the digestive tract (mainly bile salts) naturally mediate this process [17]. To improve the formation of micelles, exogenous emulsifiers are used [16]. The digestibility of fat depends on many factors, such as the age of the birds or the type and quality of the fat [18]. Studies confirm that animal fats remain largely undigested due to the predominance of saturated fatty acids in the structure of their chains [19]. Previous research has shown that undigested fat can cause enzymatic digestion disorders, which in turn translates into malabsorption of all nutrients [20]. Moreover, Danicke et al. [21] observed that the high-fat content in the digesta can affect the slowness of the passage through the intestines, which in turn can increase the microbiome activity in the ileum and thus can lead to energy loss and the development of an unfavorable microbiome. Later studies comparing the effect of various feed fats on the microbial activity in the gastrointestinal tract of birds showed that both lard and palm kernel fatty acids distillers caused a decrease in pH and an increase in total organic acid content in the ileum and caecum compared to the groups fed with soybean oil [22]. It is also known that excessive microflora development in the ileum may directly affect fat digestibility. Studies show that some groups of bacteria produce enzymes that deconjugate bile salts, resulting in poor fat emulsification and lipid absorption [23,24].

Interestingly, previous studies have shown that an additional increase in carbohydrate digestibility (NDF) can be obtained by using an emulsifier in feed containing xylanase [25]. It can be assumed that the increase in NDF digestibility is due to the improved availability of the substrate for bacterial enzymes resulting from the improvement of the fat digestibility process after adding the emulsifier. Studies on ruminants have shown that emulsifiers can improve rumen fermentation in vitro due to their direct and indirect effect on the microbiome [26,27]. These authors observed a higher microbial population in the rumen and hydrolytic enzyme activity in the rumen fluid. Based on above, it could be hypothesized that emulsifier and xylanase will stimulate the activity of chicken’s microbiome by better availability of the substrate (structural carbohydrates) for bacterial enzymes in the context of improving fat digestion.

In this study, we aimed to investigate the effect of emulsifiers, xylanase, and a combination of both in wheat diets with a high level of tallow on the gastrointestinal tract microbiota abundance and activity, excretion of total and free sialic acids, and selected gut segments morphology of broiler chickens.

## 2. Materials and Methods

All experimental procedures complied with the guidelines of the Local Ethical Committee for Experiments on Animals in Poznan regarding animal experimentation and animal care under study (European Union (EU) Directive 2010/63/EU for animal experiments). Individual approval for this trial was not required because of the production standards used during this study, and all samples were collected post-slaughter or during the study but without a negative impact on animal welfare (excreta collection).

### 2.1. Experimental Birds and Diets

The trial was performed on 480 one-day-old male ROSS 308 broiler chickens randomly allocated into 4 dietary groups: control (CON), control plus xylanase (XYL), control plus emulsifier (EMU), and control plus xylanase and emulsifier (XYL + EMU).

The experimental diets are shown in Table 1. The starter diets were based on a wheat–maize–soybean meal with rapeseed oil (as the only source of supplemental fat). Beef tallow was added to the grower and finisher diets. The amount of beef tallow varied by 4.3% in grower diets to about 6% in finisher diets. Nutrients have met or exceeded Aviagen’s recommendations for broiler chickens [28].

Diets were prepared using a horizontal mixer (Zuptor 300 MPW, Zuptor sp. Zoo., Gostyń, Poland) using the following settings—mixing time: 4 min; mixing band setting: 27.4 rpm. All feed components (except minerals, vitamins, amino acids, enzyme, emulsifier, and fat) were ground using a Skiold disk mill (SK2500, Skiold A/S. Sæby, Denmark). Non-grinding ingredients were added directly to the mixer. To obtain a homogeneous feed, the enzyme was mixed with a small amount of basic feed as a premix, which was then added to the feed to achieve final concentration. For the same reason, the emulsifier was first mixed with oils. The birds had unlimited access to water and feed in loose form. Titanium dioxide was added to the finisher diets at 3.0 g/kg, which served as an internal marker we used to determine the content of sialic acid. All diets were formulated as isoenergetic and isonitrogenic.

The treatments of microbial xylanase products given to XYL and XYL + EMU (Econase HCP 4000 ABVista, Marlborough, UK) contained 1.4-β-xylanase and the inclusion rate was 4 g per ton of feed. The emulsifier glyceryl polyethylene glycol ricinolate (GRP) (E484, Bredol 683, AkzoNobel SC AB, Stenungsund, Sweden) was added to the diets in the EMU and XYL + EMU treatments instead of wheat at a concentration of 0.15% in the starter diets, 0.173% in grower diets, and 0.188% in finisher diets.

### 2.2. Bird Management and Sample Collection

One-day-old male chicks of the Ross 308 strain were purchased at a local hatchery (DanHatch Poland, Wolsztyn, Poland). In each experimental group, there were 15 replications, each consisting of 8 birds. Before the experiment, all birds were weighed individually and assigned to 4 weight classes. Then, from each weight class, 2 birds were selected and randomly placed in pens so that the average initial body weight (BW) of birds was similar across pens. The experiment lasted 42 days.

The chicken pens were covered with wood shavings and their floor had dimensions of 1.2 by 0.9 m. Initially (up to the 7th day of study), in the experimental room, the light was on 24 h and the temperature was 32 °C, then the light program was changed to 18 h of light and 6 h of darkness, and the temperature was gradually reduced until reaching 23 °C at the end of the 3rd week of the experiment. Mortality in the experiment was low in total 4%.

To determine the content of sialic acid in excreta on the 28th and 36th day of the experiment, collecting trays were placed in the pens (after removing the birds), which allowed the collection of excreta. Samples were taken twice from 8 randomly selected pens from each experimental group (*n* = 8). Approximately 3 h after installing the collection trays, excreta without any contaminations (feed, feathers, or litter) were collected then frozen, lyophilized, and ground finely for further analysis.

On day 35, 25 chickens from each group were euthanized by electric stunning following the recommendations for the euthanasia of experimental animals. Then, the tissues needed for further analysis were collected.

The abdominal cavity was opened and the gastrointestinal tract was excised. The content of the ileum lumen (from a distance of 1 cm from the Meckel diverticulum to the ileocecal junction) and the caecum from each segment was taken from 16 birds from each group. To provide sufficient material for analysis, the collected segments were gently squeezed, and then samples from two birds were pooled into one by segments (*n* = 8). Immediately after collection, part of the ileum digesta sample was used to determine digesta viscosity.

A 2 cm segment of the middle duodenum and middle ilium was collected for histomorphometric measurements. Approximately 1.5 g of digesta from ileum and caecum was sampled and immediately placed in sterile test tubes to determine bacterial enzyme activity and then stored at −32 °C until analysis (*n* = 8). The ileal and cecal digesta samples to be used for short-chain fatty acid (SCFA) analysis were mixed with deionized water (1:1 w/w) and converted to their respective sodium salts by adjusting the pH to 8.2 with 1 M NaOH (*n* = 8).

### 2.3. Feed Analyses

Representative samples of feed were ground and passed through a 0.5 mm sieve and then analyzed (*n* = 4) for crude protein (CP), ether extract (EE) using methods 976.05 and 920.39, respectively, according to the procedure of the Association of Official Agricultural Chemists (AOAC) [29]. Gross energy (GE) was determined in the experimental diets using an adiabatic bomb calorimeter (KL 12Mn, Precyzja-Bit PPHU, Poland) that was standardized with benzoic acid. TiO_2_ was determined according to the method described by Short et al. [30], and the samples were prepared following the procedure presented by Myers et al. [31].

### 2.4. Viscosity

Ileal digesta viscosity was determined using approximately 2 g (wet weight) of the fresh digesta. After collection from the ileum, the digesta was immediately placed in a microcentrifuge tube and centrifuged at 12,700× g for 5 min. The supernatant was withdrawn and stored on ice until viscosity (mPas·s = cP = 1 × 100 dyne s cm^−2^) was determined using a Brookfield Digital DV-II+ cone/plate viscometer (Brookfield Engineering Laboratories Inc., Stoughton, MA, USA) at a shear rate of 42.5 s^−1^ at 40 °C.

### 2.5. Sialic Acid Content

First, to determine the content of total and free sialic acids following the procedure described by Jourdian et al. [32], the crude mucin was extracted from excreta according to the method described by Lien et al. [33]. To isolate the crude mucin, approx. 0.6 g freeze-dried excreta was combined with 5 mL sodium chloride (0.15 mol/l), containing 0.2. M/l sodium azide at 4 °C. Samples were homogenized and immediately centrifuged at 12,000 *g* for 30 min. The supernatant was decanted into a second test tube and centrifuged again at 12,000 *g* for 30 min to ensure the complete removal of insoluble material. An aliquot of the aqueous fraction was pipetted into a reweighed test tube and cooled in an ice bath. Ice-cold ethanol was added to a final concentration of 60% (by vol) and samples were set to precipitate overnight at −20 °C. Crude mucin was recovered by centrifugation at 1400 *g* for 10 min. The crude mucin was solubilized in 2 mL distilled water. A total of 0.1 mL of 0.04M periodic acid solution was added into 0.5 mL of crude mucin preparation. The solution was thoroughly mixed and allowed to stand in an ice bath for 20 min. After the addition of the resorcinol reagent (1.25 mL), the solution was mixed, placed in an ice bath for 5 min, heated at 100 °C for 15 min, cooled, and 1.25 mL of tert-butyl alcohol was added. Vigorous mixing yielded a single-phase solution. The tubes were placed in a 37 °C water bath for 3 min to stabilize the color, cooled to room temperature, and the absorbance was read at 630 nm using a Media spectrophotometer (Marcel Lamidey S.A., Châtillon, France). Sialic acid (total and free) in excreta was expressed in µmol/g TiO_2_.

### 2.6. Bacterial Enzyme Activity

The glycolytic activities of bacterial enzymes in the ileal and cecal digesta—α- glucosidase, β-glucosidase, α-galactosidase, β-galactosidase, and β-glucuronidase—were determined spectrophotometrically according to a previously reported protocol by Konieczka and Smulikowska [34]. The following substrates were used: p-nitrophenyl-α-D-glucopyranoside for α-glucosidase, p-nitrophenyl-β-D-glucopyranoside for β-glucosidase, p-nitrophenyl-α-D-galactopyranoside for α-galactosidase, p-nitrophenyl-β-D-galactopyranoside for β-galactosidase, and p-nitrophenyl-β-D-glucuronide for β-glucuronidase (Sigma Chemical Co., St. Louis, MO, USA).

### 2.7. Intestinal Histomorphometry

For histomorphometric measurements, the methodology previously described by Konieczka et al. [35] was used; briefly: fixed tissue samples were dehydrated, embedded in paraffin wax, and cut on a microtome into transverse sections (4.5 µm). The sections were then mounted on slides and stained with hematoxylin and eosin. Images were analyzed using light microscopy (Olympus BX51 microscope; Olympus Corp., Tokyo, Japan) and the CellD Imaging Software (Olympus Soft Imaging Solutions, Münster, Germany). Measurements were taken only from sections where the section plan ran a vertical orientation (averages represent at least two slides with a minimum of 10 well-oriented indices). The following parameters were measured: villus height (VH), measured from the tip of the villus to the villus-crypt junction; crypt depth (CD), measured from the crypt mouth to base; thickness of the intestinal wall (WT); lamina muscularis mucosae thickness; villus width (VW) taken at the midline of the villus; villus surface area (VA), calculated as villus perimeter × VH. f). The VH:CD ratio was calculated.

Neutral mucin, as an indicator of goblet cells, was assessed by staining with periodic acid–Schiff reagent according to the procedure described earlier by Rezaei et al. [36]. The number of goblet cells was calculated per 100 µm of the villus height.

### 2.8. SCFA Concentration

The SCFA determination in the ileum and caecum digesta was performed according to the procedure described previously [37], on an HP 5890 Series II gas chromatograph (Hewlett Packard, Waldbronn, Germany) with a flame-ionization detector (FID) and a Supelco Nukol fused silica capillary column (30 m × 0.25 mm internal diameter, film 0.25 mm). Helium was employed as the carrier gas. The concentrations of individual SCFAs were estimated with an internal standard (isocaproic acid) using a mixture of standard solutions.

### 2.9. Bacterial DNA Isolation

Cecal digesta samples for the isolation of bacterial genomic DNA were prepared as described by Zhu et al. [38] with some modifications. Bacterial genomic DNA was extracted from digesta using the QIAamp Fast DNA Stool Mini Kit (Qiagen, Stockach, Germany) according to the manufacturer’s protocol. Then, the yield and purity of the isolated DNA were estimated spectrophotometrically (Nanodrop, NanoDrop Technologies, Wilmington, DE, USA).

### 2.10. Polymerase Chain Reaction (PCR) Amplification of Bacterial 16S rRNA Gene

The primers and PCR conditions used to amplify the bacterial 16S rRNA gene are shown in Table 2. The universal primer set was used to determine the total bacteria population. The PCR conditions were applied as reported above for each respective bacteria group. The obtained PCR-products were separated by electrophoresis on a 2% agarose gel. PCR products were quantified using ImageJ 1.47v software for densitometry measurements (National Institute of Mental Health, Bethesda, MD, USA), with a density of bands for each bacteria group expressed in relation to the density of the total bacteria primers product. Each sample was analyzed in duplicate.

### 2.11. Calculations and Statistical Analyses

The surface of the villi was calculated according to the formula (Equation (1)) given by Sakamoto et al. [39]:(1)Villus surface= (2π)×(VW/2)×(VH),
where VW = villi width and VH = villi height.

For sialic acid content and digesta viscosity, a single pen (*n* = 15) was considered a replicate unit for the statistical analysis. For the analysis of other parameters, individual birds (*n* = 8) were considered experimental units. Data variability was expressed as a pooled standard error of the mean (SEM), and *p* < 0.05 was considered statistically significant. The experiment was completely randomized and all data were subject to analysis of variance using the general linear model procedure of the R environment [40] and using the Agricolae package [41] according to the following model (Equation (2)):(2)$$Y_{i}=\mu +\alpha_{i}+\varepsilon_{i},$$
where *Y_i_*—the measured dependent variable; *µ*—overall mean; *α_i_*—the effect of feed additives; *ε_i_*—random error.

## 3. Results

### 3.1. Excretion of Sialic Acid in Ileum and Viscosity of Digesta

There was no effect of the additives on the excretion of sialic acid (total and free) on both the 28th and 35th day of the experiment (Table 3). The use of XYL and a combination of XYL + EMU had a statistical effect on reducing the viscosity of the digesta. The viscosity value of the XYL treatment differed from the CON by 1.17 cP and from that of the XYL + EMU treatment by 1.72 cP.

### 3.2. Bacterial Enzyme Activity

The activities of all five bacterial enzymes (Table 4) in the ileal digesta were generally lower in birds fed experimental diets compared to the CON group. In three of them, a statistically significant difference was noted. In the case of α-galactosidase (*p* < 0.0001) and α-glucosidase (*p* < 0.001), the decrease in enzyme activity was the same in all experimental groups, whereas in β-glucosidase in the EMU and XYL + EMU groups, a greater decrease was observed than in the case of the addition of xylanase alone (*p* < 0.0001). In the case of the cecum, bacterial enzyme activity differed statistically significantly between the groups only in the case of β-glucosidase (*p* < 0.0001) and β-glucuronidase (*p* < 0.05). In both cases, the simultaneous use of xylanase and emulsifier increased enzymatic activity. Enzyme activity in the cecum was significantly higher than in the ileum.

### 3.3. Histomorphometry of Duodenum and Ileum

The effect of investigated additives on morphological parameters of the duodenum and ileum in 35 day old birds is shown in Table 5. The ratio of VA and VH:CD data were not shown due to the lack of statistical differences. In the duodenum, only changes in the number of goblet cells were noted. Each of the experimental additives used resulted in a decrease in the number of goblet cells on the villi (*p* < 0.05). In the XYL + EMU treatments, the most significant difference was noted from 12 in CON treatment to 9.8. In the ileum, an increase in the depth of crypts was noted after the application of both additives (*p* < 0.05). The simultaneous use of XYL + EMU deepened the crypts by more than 20 µm.

### 3.4. SCFA Concentration in Digesta

There were no significant differences in total SCFA concentration in ileal content (Table 6). Only the concentration of valerate statistically increased in the XYL + EMU group to 0.116 from 0.068 µmol/g in the CON group (*p* < 0.01). The total SCFA concentration in the cecal digesta was lowest in birds from the CON group (*p* < 0.01). The increase in the total SCFA concentration after application of additives results directly from the increase in acetate (*p* < 0.01) and butyrate (*p* < 0.0001) concentration, there was no difference whether XYL or EMU were used together or separately (Table 7).

### 3.5. Analysis of Selected Microbiota Abundance

Table 8 shows that the relative abundance of *Bifidobacterium*, *Lactobacillus,* and *Escherichia coli* did not change after using either the xylanase or emulsifier. The relative abundance of *Clostridium* was significantly (*p* < 0.05) lower in groups with emulsifier addition (EMU and XYL + EMU).

## 4. Discussion

The suitability of feed additives for use in poultry nutrition can be determined by using classic indicators such as performance or digestibility of nutrients. Determining the effect of feed additives on morphology and digestive tract physiology as well as on the composition and activity of gut microflora is becoming increasingly popular. It is well known that the intestinal microflora has a huge impact on the growth and health of broilers and that it is possible to change the composition and activity of the microflora by manipulating their diet [20].

Studies on the effect of soluble NSP and NSP degrading enzymes on the morphology of the birds’ gastrointestinal tract are ambiguous and the changes are difficult to explain. In the present study, the independent use of xylanase and emulsifier in wheat-based diets had the same effect on the reduction of the number of goblet cells in the duodenum and the deepening of crypts in the mucosa of the ileum of birds, although, in the XYL + EMU treatment, an additive effect was noted (Table 5). These results partly agree with earlier studies in which the authors did not note the effect of XYL on the gastrointestinal morphometry of birds fed wheat-based diets [42]; however, later studies showed an increase in goblet cell number in the duodenum epithelium after XYL addition [43]. In addition, the authors of the above-mentioned papers did not notice changes in the depth of the crypts in the ileum. Interestingly, in other studies where the jejunum epithelium was examined, crypt deepening was also noted after XYL supplementation [44]. Deeper crypts indicate a greater activity in these regions of the intestinal epithelium (faster cell renewal, increased enterocyte exchange, and a greater demand for tissue development) [45]. There are goblet cells in the crypts, which are responsible, among others, for mucus production—it can be assumed that the deepening of crypts may be correlated with an increase in mucus production [46]; however, in the present study, it appears that the increase in goblet cells was not related to sialic acid secretion. Interestingly, on the 28th day of the experiment, a tendency to reduce the presence of sialic acid after the use of additives in the ileum was observed (Table 3).

There are no reports in the literature on the effect of dietary fat and emulsifiers on gastrointestinal morphology, so it is difficult to explain the results obtained. The reduction in the villi goblet cell number in the duodenum after using the emulsifier may result from an increase in the fat digestibility and faster passage of content, which may be associated with a reduced demand for mucus production produced by goblet cells.

Table 3 shows that there was a trend (*p* < 0.01) of the XYL and EMU on the content of sialic acid in excreta at day 28th but not at day 35th. Sialic acid is one of the components of mucins that cover the epithelium of the gastrointestinal tract and its amount in the excreta is considered a good indicator of endogenous losses [47]. Mucins are a substrate for intestinal bacteria that convert them into free sialic acid. By determining the content of sialic acid in excreta, it is possible to obtain information about the total production of mucins in the digestive tract [48]. Earlier studies show that some fiber fractions, including those found in wheat, have an impact on the increase in endogenous losses similarly to antinutritional substances such as alkaloids, phytates, or NSP [49] but also some fiber fractions, including those found in wheat [4] have an impact on the increase in endogenous losses. In addition, it is thought that mucin production may be affected by bacterial infections and epithelial damage caused by bacterial toxins [50]. Our results show that the addition of xylanase and emulsifier did not reduce endogenous losses. These results do not coincide with previous ones in which a decrease in endogenous losses was noted after using NSP degrading enzymes [50,51].

To determine the effect of feed additives on the gut microflora activity, it is worth looking at the concentration of SCFA and bacterial enzyme activity. It is known that NSP contained in wheat and the high content of beef tallow negatively affect the symbiotic microflora and may cause an increase in the population of pathogenic bacteria [21,52]. Disruption of the normal functioning of the digestive tract (by increasing the viscosity or extending the passage of digesta) may contribute to an increase in the activity of microbial fermentation in the lower sections of gastrointestinal tract (GIT), which in turn reduces the efficiency of enzymatic digestion and causes energy losses for the host [53]. Studies show that, by using exogenous enzymes, microbial populations in the digestive tract can be indirectly affected by increasing the total pool of substrates that bacteria use as a carbon source [13,54].

In the present study, SCFA concentration was detected in the ileum and cecum (Table 6 and Table 7). The cecum is the main location of microbial fermentation and the digesta stays longer than in other GIT sections, consequently, SCFA concentration is higher than in the ileum [37,55]; our study confirmed this finding.

After using both xylanase and emulsifier, there were no statistical differences in the total concentration of SCFA in the ileum and it was only a slight increase in valerate in the XYL + EMU group (Table 6). The results obtained are not consistent with previous studies, which report that after the use of enzymes that improve the digestibility of nutrients, the activity of microflora in the ileum should decrease due to the limited amount of substrate available for microbial fermentation [56]; however, our results are consistent with those obtained by Józefiak et al. [57], who also did not notice the effect of xylanase on the reduction of microbial activity in the ileum.

On the other hand, there was a significant increase in total SCFA concentration in the cecum after adding XYL and EMU (Table 7), which is in agreement with studies of Choct et al. [13] and Jamroz et al. [58]. It can be assumed that XYL works according to the theory presented by Bedford and Patridge [59], which says that the action of feed enzymes can be divided into two phases: the ileum and the cecum. In the first phase, xylanase and emulsifier reduce the viscosity of the digesta. In the cecum phase, the oligosaccharides (xylose and xylo-oligomers) formed due to the degradation of NSP become an additional substrate for bacteria residing in the cecum, thus stimulating the production of SCFA [4,60]. Thus, we speculate that the XYL treatment in the present study resulted in the NSP-fractions of the diet being fractioned to the chain length able to enter the caecum where it served as substrates for bacteria fermentation, and additional EMU supplementation supported this process. Fermentation at the caecum does not cause losses of nutrients but also provides additional (although not high) amounts of energy that can be used by the host [61]. SCFA produced in the cecum also affects microbiome regulation and intestinal epithelial development [55].

Both XYL and EMU increased total SCFA concentration. They particularly affected the increase in acetate and butyrate production. Butyrate is a good source of energy for enterocytes located on the surface of the intestinal epithelium. In addition, butyric acid regulates mucin production, participates in immune processes, and down-regulates the development of pathogenic microflora in the gut. The high content of acetate reduces the pH in the cecum, which prevents the development of pathogenic microflora [62]. It seems that the changes in the *Clostridium spp*. population observed in this trial could be caused by higher butyrate and acetate production after XYL or EMU application.

Table 4 shows the results of bacterial enzyme activity in the ileum and cecum. As mentioned above, soluble NSP contained in wheat negatively affect digestive processes and lengthen the passage time of digesta. Undigested nutrients stay in ileum for a long time and can affect the composition and activity of the microflora [63,64]. Hübener et al. [7] presented studies in which an increase in the number of glycolytic bacteria in a similar situation was observed. It could be speculated that in the current study, the addition of XYL counteracted to some extent the negative effect of NSP on the viscosity and digestibility of carbohydrates caused a decrease in the activity of bacterial enzymes by lowering the amount of substrate for them. The α-galactosidase is the enzyme responsible for the hydrolyses of α-galactosides contained in oligosaccharides to galactose and sucrose [65] and α-glucosidase and β-glucosidase are responsible for the breakdown β-glucans and cellulose [66]. In the cecum, the effect of the additives used was much smaller. Only the combined use of XYL + EMU significantly influenced the increase in β-glucosidase activity (*p* < 0.0001) and to a small extent on the increase in β-glucuronidase activity (*p* < 0.05). β-glucosidase is an enzyme that hydrolyses some NSPs and ferments starch [67,68]. It seems that the use of xylanase release some of the nutrients providing substrates for fermentation by bacteria; however, the increase in β-glucuronidase activity is interesting. Studies show that the high activity of this enzyme leads to negative changes in the intestinal microflora, which is indicative of the population of *E. coli* and *Clostridium* growth [69,70]. During the experiment, the mortality of birds did not differ from the average and during the sampling, no areas affected by disease in the gastrointestinal tract were noticed.

The results of the current study indicate that the addition of EMU affects the quantitative and qualitative composition of the selected gut microbiota. After using the EMU, the activity of bacterial enzymes in the small intestine decreased significantly (Table 4) and the concentration of SCFA in the cecum increased (Table 7). According to the authors’ knowledge, there are no reports in the literature about the impact of dietary fat and the use of EMU on the gut microbiota in poultry; therefore, the explanation of these phenomena can be based on the known mechanisms-of-action of EMU. Based on available research describing the impact of the addition of EMU on the production results, we assume that the effect of EMU on the gut microflora in birds is indirect. Throughout the improvement of fat digestibility [25], which physically blocked the access of digestive enzymes to nutrients breakdown and slow down the passage of digesta [21], the bacterial enzyme activity was depressed. This may be due to the accelerated passage of digesta through the ileum [16]; the bacteria had too little time to produce enzymes. By reducing the activity of bacterial enzymes, the host was able to use more feed ingredients through enzymatic digestion, which can improve production results. This assumption is confirmed by earlier studies [25] where an improvement in apparent total tract digestibility of ether extract and production results (mainly in the grower period) was obtained using a similar experimental design. After using EMU, the digesta content of fat entering the cecum was reduced to a high extent (ATTD of EE is ca. 89%) and contains carbohydrate fractions (NSP, resistant starch), which undergo microbial fermentation. Substrates delivered to the cecum are willingly used by microorganisms and the effect is an increase in SCFA concentration.

The addition of EMU reduced the abundance of *Clostridium* spp. in the cecum (*p* < 0.05) (Table 8). Earlier studies by Józefiak et al. [22] have shown that dietary fat may affect the composition and activity of microflora in the gastrointestinal tract of birds. The authors suggest that this is an indirect action. Beef tallow was used in our experiment, which contains high levels of saturated fat—causing impeded digestion in birds. The high-fat content in the feed can lead to a significant extension of the passage of digesta and reduction of digestive processes in the ileum of the chickens [20]. This in turn may contribute to an increased shift in abundance of bacteria from the *Clostridium* spp. in the caecum. The addition of EMU that improves the digestibility of fat and other nutrients, thus accelerating the passage of digesta, contributed to reducing *Clostridium* abundance.

Interestingly, the addition of an emulsifier stimulated butyrate secretion in the cecum, which is produced, among others, by bacteria of the genus *Clostridium*. *Clostridia* in the chicken gut occur mainly in the caecum and their large number does not necessarily indicate bird health abnormalities and compromised production results. The authors of other studies suggest that the majority of *Clostridial* bacteria do not cause any loss to the host and can be beneficial for them, e.g., *Clostridium butyricum*, which belongs to this group and is a butyric acid-producing bacteria beneficial to the host [71]. This may indicate that by improving the conditions in the lumen of the intestine, the emulsifier reduced the total number of bacteria of the genus Clostridia, but did not significantly affect those bacteria that produce butyrate, and it seems that it positively influenced their activity. One of the clinical signs of *Clostridial* enteric disease includes thin-walled and friable intestines [72]. In this regard, no negative influence of feeding broilers diet supplemented EMU was seen in the gut mucosa of the birds, which was manifested in uncompromised wall thickness (data did not differ significantly between all groups). It seems that these findings also confirm that the EMU additive did not evoke the shift between beneficial and pathogenic to the host *Clostridia* abundance in the gut.

## 5. Conclusions

In conclusion, the use of EMU and XYL in wheat-based diets with beef tallow used as a supplemental fat for poultry reduces the microbiota activity in the ileum and increases its activity in the cecum. As a result, enzymatic digestion dominates in the duodenum and ileum, and enhanced microbial fermentation takes place only at the caecum, which allows the host to make optimal use of nutrients. It seems that using both additives exerts a cumulative effect to some extent on gut microbiota activity.

## Figures and Tables

**Table 1 animals-10-02197-t001:** Composition and nutrient contents of the experimental diets: starter, grower, and finisher (as feed basis).

Ingredients [%]	1–11 d	11–25 d	25–42 d
Wheat	20	58.53	57.42
Maize	31.77	-	-
SBM	37.56	28.83	28.43
Rapeseed oil	6.52	4.30	3.20
Tallow	-	4.36	6.19
Monocalcium phosphate	1.42	1.29	1.20
Premix ^1^	1	1	1
Limestone	0.52	0.44	1.17
DL-met	0.36	0.23	0.27
HCL-Lys	0.29	0.22	0.22
NaHCO_3_	0.25	0.18	0.4
NaCl	0.21	0.24	0.11
L-Thr	0.12	0.07	0.09
TiO_2_	-	0.30	0.30
Analyzed nutrient content			
Crude protein [%]	22.94	20.45	20.92
Gross energy [MJ/kg]	18.23	18.43	18.57
Ether extract [%]	4.24	9.29	9.86
TiO_2_ [%]	-	0.28	0.28

^1^ Provides per kg diet: IU: vit. A 11250. cholecalciferol 2500; mg: vit. E 80, menadione 2.50, vit. B12 0.02, folic acid 1.17, choline 379, D-pantothenic acid 12.5, riboflavin 7.0, niacin 41.67, thiamin 2.17, D-biotin 0.18, pyridoxine 4.0, ethoxyquin 0,09. Mn 73, Zn 55, Fe 45, Cu 20, I 0.62, Se 0.3, salinomycin 60.

**Table 2 animals-10-02197-t002:** Sequences of primers used for amplification of bacterial 16S rRNA gene.

Bacterial Group	Primers	Sequence 5′-3′	Base Pair
Universal	Forward	CGTGCCAGCCGCGGTAATACG	611
	Reverse	GGGTTGCGCTCGTTGCGGGACTTAACCCAACAT	
*Lactobacillus* spp.	Forward	CATCCAGTGCAAACCTAAGAG	286
	Reverse	GATCCGCTTGCCTTCGCA	
*Escherichia coli*	Forward	GGGAGTAAAGTTAATACCTTTGCTC	585
	Reverse	TTCCCGAAGGCACATTCT	
*Clostridium* spp.	Forward	AAAGGAAGATTAATACCGCATAA	722
	Reverse	ATCTTGCGACCGTACTCCCC	
*Bifidobacterium* spp.	Forward	CGGGTGCTICCCACTTTCATG	1417
	Reverse	GATTCTGGCTCAGGATGAACG	

**Table 3 animals-10-02197-t003:** Excretion of sialic acid in ileum and viscosity of digesta in birds fed control (CON), control supplemented with xylanase (XYL), control supplemented with an emulsifier (EMU), and control diet supplemented with xylanase and emulsifier XYL + EMU.

Dietary Treatments	Sialic Acid [µm/g TiO_2_]				Viscosity (cP)
	Total (at 28 d)	Free (at 28 d)	Total (at 35 d)	Free (at 35 d)	
CON	716	631	766	683	3.69 ^a^*
XYL	556	546	621	584	2.52 ^b^
EMU	665	521	720	579	3.23 ^a^
XYL + EMU	551	444	607	501	1.97 ^c^
SEM	32.85	32.14	49.22	45.03	0.113
*p*	0.076	0.055	0.401	0.348	<0.001

* Mean in a column not sharing a common letter (a–c) are statistically different (*p* ≤ 0.05); SEM—pooled standard error of mean.

**Table 4 animals-10-02197-t004:** The activity of the bacterial enzyme [U/g] ^1^ in ileal and cecal digesta in birds fed control (CON), control supplemented with xylanase (XYL), control supplemented with an emulsifier (EMU), and control diet supplemented with xylanase and emulsifier XYL + EMU.

Dietary Treatments	Ileum					Caecum				
	Galactosidase		Glucosidase		β-Glucuronidase	Galactosidase		Glucosidase		β-Glucuronidase
	α-	β-	α-	β-		α-	β-	α-	β-	
CON	0.192 ^a^*	0.178	0.059 ^a^	0.054 ^a^	0.057	1.190	2.272	0.544	0.359 ^b^	1.256 ^ab^
XYL	0.137 ^b^	0.176	0.056 ^b^	0.046 ^b^	0.059	0.904	2.00	0.472	0.331 ^b^	1.121 ^ab^
EMU	0.138 ^b^	0.173	0.045 ^b^	0.038 ^c^	0.059	1.252	2.253	0.537	0.373 ^b^	1.232 ^ab^
XYL + EMU	0.119 ^b^	0.173	0.047 ^b^	0.038 ^c^	0.055	1.316	2.312	0.597	0.497 ^a^	1.478 ^a^
SEM	0.0068	0.0041	<0.001	0.0017	0.00071	0.0529	0.0848	0.0151	0.0155	0.0442
*p*	<0.0001	0.616	<0.001	<0.0001	0.425	0.128	0.63	0.99	<0.0001	<0.05

^1^ U = μmol of p-(o-) nitrophenol formed per min per g of a sample; * means in a column not sharing a common letter (a–c) are significantly different (*p* ≤ 0.05); SEM—pooled standard error of mean.

**Table 5 animals-10-02197-t005:** Morphometry of duodenum and ileum mucosa in birds fed control (CON), control supplemented with xylanase (XYL), control supplemented with an emulsifier (EMU), and control diet supplemented with xylanase and emulsifier XYL + EMU.

Dietary Treatments	Duodenum Mucosa				Ileum Mucosa			
	Villus Height	Villus Width	Crypt Depth	Goblet Cells	Villus Height	Villus Width	Crypt Depth	Goblet Cells
	[µm]				[µm]			
CON	1216	112	162	12.00 ^a^*	714	105	138 ^b^	2.37
XYL	1360	118	146	11.20 ^ab^	880	112	148 ^ab^	2.03
EMU	1283	120	150	10.10 ^ab^	735	105	144 ^ab^	2.10
XYL + EMU	1262	113	155	9.80 ^b^	795	108	159 ^a^	2.01
SEM	28.70	2.02	3.61	0.349	34.90	1.50	2.85	0.068
*p*	0.803	0.804	0.543	<0.05	0.531	0.823	<0.05	0.103

* Mean in a column not sharing a common letter (a,b) are statistically different (*p* ≤ 0.05); SEM—pooled standard error of mean.

**Table 6 animals-10-02197-t006:** Total short-chain fatty acid (SCFA) concentration in ileum digesta in birds fed control (CON), control supplemented with xylanase (XYL), control supplemented with an emulsifier (EMU), and control diet supplemented with xylanase and emulsifier XYL + EMU.

Dietary Treatments	Ileum (µmol/g)						
	Acetate	Propionate	Isobutyrate	Butyrate	Isovalerate	Valerate	Total SCFA **
CON	5.50	0.286	0.145	0.308	0.071	0.068 ^b^*	6.37
ENZ	6.07	0.328	0.163	0.194	0.117	0.081 ^ab^	6.95
EMU	5.92	0.292	0.163	0.112	0.074	0.104 ^ab^	6.67
ENZ + EMU	5.18	0.262	0.145	0.284	0.096	0.116 ^a^	6.46
SEM	1.54	0.011	0.004	0.044	0.0066	0.0078	2.275
*p*	0.553	0.286	0.998	0.698	0.571	<0.01	0.992

* Mean in a column not sharing a common letter (a–c) are statistically different (*p* ≤ 0.05); SEM—pooled standard error of mean; ** total SCFA is the sum of the individual SCFA from the table.

**Table 7 animals-10-02197-t007:** Total short-chain fatty acid (SCFA) concentration in cecal digesta in birds fed control (CON), control supplemented with xylanase (XYL), control supplemented with an emulsifier (EMU), and control diet supplemented with xylanase and emulsifier XYL + EMU.

Dietary Treatments	Caeca (µmol/g)						
	Acetate	Propionate	Isobutyrate	Butyrate	Isovalerate	Valerate	Total SCFA **
CON	27.00 ^b^*	1.74	0.516	7.03 ^c^	0.392	0.743	37.40 ^b^
ENZ	40.40 ^a^	1.74	0.493	12.60 ^ab^	0.372	0.737	56.30 ^a^
EMU	37.90 ^a^	1.61	0.466	11.40 ^b^	0.387	0.742	52.50 ^a^
ENZ + EMU	40.30 ^a^	1.64	0.449	15.60 ^a^	0.312	0.825	59.10 ^a^
SEM	1.54	0.065	0.021	0.767	0.024	0.0245	2.275
*p*	<0.01	0.465	0.185	<0.0001	0.306	0.225	<0.01

* Mean in a column not sharing a common letter (a–c) are statistically different (*p* ≤ 0.05); SEM—pooled standard error of mean; ** total SCFA is the sum of the individual SCFA from the table.

**Table 8 animals-10-02197-t008:** Genera- and species-level bacterial populations from cecal contents measured on the 35th day of life in birds fed control (CON), control supplemented with xylanase (XYL), control supplemented with an emulsifier (EMU), and control diet supplemented with xylanase and emulsifier XYL + EMU.

Dietary Treatments	*Bifidobacterium* spp.	*Lactobacillus* spp.	*E. coli*	*Clostridium* spp.
CON	0.7474	0.7086	0.0866	1.037 ^a^*
XYL	0.7414	0.6246	0.1265	0.9934 ^a^
EMU	0.5037	0.6034	0.0756	0.8471 ^b^
XYL + EMU	0.7195	0.6399	0.077	0.8134 ^b^
SEM	0.0485	0.0346	0.0184	0.0452
*p*	0.487	0.502	0.612	<0.05

* Mean in a column not sharing a common letter (a,b) are statistically different (*p* ≤ 0.05); SEM—pooled standard error of mean.

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
