# Peer review of "Emulsifier and Xylanase Can Modulate the Gut Microbiota Activity of Broiler Chickens"

_animals, 2020, doi:10.3390/ani10122197_

Round 1

Reviewer 1 Report

The paper presents the effect of emulsifiers, xylanase, and a combination of both on the gastrointestinal tract microbiota abundance and activity, excretion of total and free sialic acids, and selected gut segments morphology of broiler chickens fed in wheat-based diets with beef tallow.
There was observed that these supplements reduce the microbiota activity in the ileum and increases its activity in the cecum. As a result, enzymatic digestion dominates in the duodenum and ileum and enhanced microbial fermentation takes place only at the caecum. It seems that using both additives exerts a cumulative effect on gut microbiota activity and allows to optimal use of nutrients by chicken broilers.

The article is interesting and the results are clearly presented. TIn my opinion the manuscript needs the minor revisions only. I have only a general note for the discussion chapter i.e. Paragraphs for discussion should begin with a recall of experiment result, and only then an explanation of it. In the reviewed manuscript, the authors very often discuss the problem first and finally refer to the results obtained. This arrangement of the text often makes it difficult to understand it fully.

I showed more comments in the manuscript.

Author Response

The authors would like to thank the reviewers and Editor for a helpful critique of the manuscript. The manuscript has been improved to reflect the required changes and these modifications are detailed below and directly in text using track-changes mode.

Please add the References.

LINE 59 moved references from line 58 to line 59 afterwords „up to 80% of the total production costs”

Add the abbreviation "NSP" that is used in L65

LINE 64 changed to ” significant amounts of non-starch polysaccharides (mainly arabinoxylans)” to “significant amounts of non-starch polysaccharides (NSPs) - mainly arabinoxylans,”

L86-L88: If possible, please simplify the multi-dependent clause of this sentence.

LINE 86-88  changed “Although there is little research on the effect of dietary fat on the bird microbiome, Danicke et al. [21] observed that the high-fat content in the digesta can affect the slowness of the passage through the intestines, which in turn can increase the microbiome activity in the ileum and thus can lead to energy loss and the development of an unfavourable microbiome.” To “Moreover Danicke et al. [21] observed that the high-fat content in the digesta can affect the slowness of the passage through the intestines which in turn can increase the microbiome activity in the ileum - thus can lead to energy loss and the development of an unfavorable microbiome.”

Form. Repetition of the word "Interestingly" in the next two sentences.

Line 100 we removed “interestingly”

Please re-edit the paragraph to strengthen the justification for the research hypothesis. Can improving rumen fermentation by emulsifiers be transferred to monogastric of poultry?

Yes we believe that mode of action is the same, we changed sentences and now we have: Studies on ruminants have shown that emulsifiers can improve rumen fermentation in vitro due to their direct and indirect effect on the microbiome [26,27]. These authors observed a higher microbial population in the rumen and hydrolytic enzyme activity in the rumen fluid. Based on above it could be hypothesized that emulsifier and xylanase will  stimulate activity of chicken’s  microbiome by better availability of the substrate (structural carbohydrates) for bacterial enzymes in the context of improving fat digestion.

In this study, we aimed to investigate the effect of emulsifiers, xylanase, and a combination of both in wheat diets with the high level of tallow on the gastrointestinal tract microbiota abundance and activity, excretion of total and free sialic acids and selected gut segments morphology of broiler chickens.

Reviewer 2 Report

This is an excellent trial with very interesting findings. However, the paper lacks in clarity when it comes to stating the hypothesis (challenging diet: do additive improve/prevent negative effects?). The one major flaw is the statistical analysis with a 1-way ANOVA instead of a 2-way ANOVA. The trial design is clearly aiming at 2 main effects and therefore the statistical analysis should reflect the trial design. Most significant findings will likely be confirmed with a reanalysis but the interpretation will be simplified. In some cases I expect improved statistical power supporting many of the potential claims the authors made. 

  • The hypothesis needs to be made clearer. It should be made clearer that a diet with relatively high levels of animal fat and fiber is used. Therefore the hypothesis is that the control diet is detrimentous to the birds and that the additives relief this deleterious effect. The aim of the trial was clearly not to just randomly check if enzymes and emulsifiers have an effect on the listed parameters in a standard european diet but rather a dedicated trial design/diet formulation was used to investigate the potential usefulness of the additives in a challlenging dietary environment.
  • The clearly formulated hypothesis needs to be stated in the abstract/summary/end of introduction before a description of what was measured is given.
  • in the method section it is not stated clearly for how long the birds were raised Only stated in diet table not in text). Please add this info.
  • Otherwise excellent detailed method description.
  • Was there any performance data recorded? I understand that interpretation might be difficult due to bird sacrifices but it would help the interpretation a lot to understand if performance was in line with breeder guidelines or not. Also of course, if there were treatment differences is of interest.
  • Trial was designed as 2-factorial and should be analysed with a 2-way ANOVA with the 2 main effect XYL and EMU. There seem to be additive effects but no interactions, so a 1-way ANOVA as presented is not as useful. Clear main effect description will also make the interpretation easier and more straight forward. if a 2-way ANOVA is not used an explanation should be given in the method section as to why. IMO this is a major flaw in the analysis of an otherwise excellent trial.
  • Passage 373-385 is worded too strongly. There are trends visbile (P<0.1) and I suspect if a main effect analysis would be used XYL would give a significant results on reduction of siliac acid supporting literature evidence on reduction of endogenous loss with xylanase. While I normally appreciate not overinterpreting trends, in this case I believe the presented data in combination with literature does not warrant wordings like "our results show that the addition of xylanase and emulsifier did not reduce endogenous losses".
  • Passage 398-404: Please do not discuss results which are not shown in the results section, especially when they contain significant results. Please show the results, they are relevant.
  • It would be helpful to discuss SCFA and bacterial enzyme activity together in one paragraph. Bacterial enzyme acctivity in the Ileum is reduced by additives and changes on SCFA might be masked by absorption through the bird. Found in Passage 446-464. The passages describing the individual effects could be shortened, the discussion is a bit lengthy.
  • Line 441-442: Increase in activity does not mean high activity. Is the enzyme activity in the referenced paper on the same level as in this dataset or much higher? Please only speculate about infection if the activity is similar, as otherwise you do not have further evidence substantiating the idea, as you also do describe.
  • Line 481: Sentence is a bit mixed up. PLease correct.

Author Response

The authors would like to thank the reviewers and Editor for a helpful critique of the manuscript. The manuscript has been improved to reflect the required changes and these modifications are detailed below and directly in text using track-changes mode.

The hypothesis needs to be made clearer. It should be made clearer that a diet with relatively high levels of animal fat and fiber is used. Therefore the hypothesis is that the control diet is detrimentous to the birds and that the additives relief this deleterious effect. The aim of the trial was clearly not to just randomly check if enzymes and emulsifiers have an effect on the listed parameters in a standard european diet but rather a dedicated trial design/diet formulation was used to investigate the potential usefulness of the additives in a challlenging dietary environment.

The clearly formulated hypothesis needs to be stated in the abstract/summary/end of introduction before a description of what was measured is given

We have changed this and now we have:

Studies on ruminants have shown that emulsifiers can improve rumen fermentation in vitro due to their direct and indirect effect on the microbiome [26,27]. These authors observed a higher microbial population in the rumen and hydrolytic enzyme activity in the rumen fluid. Based on above it could be hypothesized that emulsifier and xylanase will  stimulate activity of chicken’s  microbiome by better availability of the substrate (structural carbohydrates) for bacterial enzymes in the context of improving fat digestion.

In this study, we aimed to investigate the effect of emulsifiers, xylanase, and a combination of both in wheat diets with high level of tallow on the gastrointestinal tract microbiota abundance and activity, excretion of total and free sialic acids, and selected gut segments morphology of broiler chickens.

Line 33 added “in wheat diets with a high level of tallow”

Line 39 added “in wheat diets with a high level of tallow”

line 103 changed “In this study, we aimed to investigate the effect of emulsifiers, xylanase, and a combination of both on the gastrointestinal tract microbiota abundance and activity, excretion of total and free sialic acids, and selected gut segments morphology of broiler chicken” to “In this study, we aimed to investigate the effect of emulsifiers, xylanase, and a combination of both in wheat diets with a high level of tallow on the gastrointestinal tract microbiota abundance and activity, excretion of total and free sialic acids, and selected gut segments morphology of broiler chickens.”

in the method section it is not stated clearly for how long the birds were raised Only stated in diet table not in text). Please add this info.

LINE 149 added,” The experiment lasted 42 days.”

Passage 398-404: Please do not discuss results which are not shown in the results section, especially when they contain significant results. Please show the results, they are relevant.

Line 319 moved from 324 sentences “There were no significant differences in total SCFA concentration in ileal content (Table 6). Only the concentration of valerate statistically increased in the XYL + EMU group to 0.116 from 0.068 µmol/g in the CON group (P < 0.01).”

Line 328 added table “Total short-chain fatty acid (SCFA) concentration in ileum digesta in birds fed control (CON), control supplemented with xylanase (XYL), control supplemented with an emulsifier (EMU), and control diet supplemented with xylanase and emulsifier XYL + EMU.”

Line 334 changed table number from 6 to 7

Line 346 changed table number from 7 to 8

Line 481: Sentence is a bit mixed up. PLease correct.

Line 480-482 changed ” This may indicate that by improving the conditions in the lumen of the intestine, the emulsifier reduced the abundance of bacteria from the genus Clostridia but not affected markedly of that producing butyrate  and positively affected the activity of those producing butyrate” to “This may indicate that by improving the conditions in the lumen of the intestine, the emulsifier reduced the total number of bacteria of the genus Clostridia, but did not significantly affect those bacteria that produce butyrate, and it seems that it positively influenced their activity.”

The trial was designed as 2-factorial and should be analysed with a 2-way ANOVA with the 2 main effects XYL and EMU. There seem to be additive effects but no interactions, so a 1-way ANOVA as presented is not as useful. Clear main effect description will also make the interpretation easier and more straight forward. if a 2-way ANOVA is not used an explanation should be given in the method section as to why. IMO this is a major flaw in the analysis of an otherwise excellent trial.

We used one-way ANOVA and we don’t see the possibility to use 2-way, in this study we have only one independent variable– an emulsifier, xylanase or mix – defined as “additive”. According to your suggestion, we should analyse with 2-way-ANOVA with two main effects XYL and EMU so where CON and XYL+EMU data will be presented? as a XYL or EMU ? 2-way ANOVA requires two independent variables we have only one - “additive”. Two-way ANOVA in this study is wrong in our opinion, it is impossible to present like that, this data. If we understand your idea correctly, we should pool XYL and XYL+EMU data as a one mean and as a secund mean EMU and XYL+EMU. So XYL+EMU will be used twice and CON will be excluded? that is a very rare way of data presenting. Please see our previous paper (exactly the same study design;  Kaczmarek, S. A., A. Barri, M. Hejdysz, and A. Rutkowski. 2016. “Effect of Different Doses of Coated Butyric Acid on Growth Performance and Energy Utilization in Broilers.” Poultry Science 95(4): 851–59.)

Was there any performance data recorded? I understand that interpretation might be difficult due to bird sacrifices but it would help the interpretation a lot to understand if performance was in line with breeder guidelines or not. Also of course, if there were treatment differences is of interest.

Performance results and digestibility data are subject of the next/another paper.

Passage 373-385 is worded too strongly. There are trends visbile (P<0.1) and I suspect if a main effect analysis would be used XYL would give a significant results on reduction of siliac acid supporting literature evidence on reduction of endogenous loss with xylanase. While I normally appreciate not overinterpreting trends, in this case I believe the presented data in combination with literature does not warrant wordings like "our results show that the addition of xylanase and emulsifier did not reduce endogenous losses".

We have changed this sentence and now we have: Table 3 shows that there was a trend (P<0.01) of the XYL and EMU on the content of sialic acid in excreta at day 28th but not at day 35th.

Found in Passage 446-464. The passages describing the individual effects could be shortened, the discussion is a bit lengthy.

We made some changes and now paragraph is shorter:

Table 4 shows the results of bacterial enzyme activity in the ileum and cecum. As mentioned above, soluble NSP contained in wheat negatively affect digestive processes and lengthen the passage time of digesta. Undigested nutrients stay in ileum for a long time and can affect the composition and activity of the microflora [63, 64]. Hübener et al. [7] presented studies in which an increase in the number of glycolytic bacteria in a similar situation was observed. In the current study, the addition of XYL counteracted to some extent the negative effect of NSP on the viscosity and digestibility of carbohydrates caused a decrease in the activity of bacterial enzymes by lowering the amount of substrate for them. The α-galactosidase is the enzyme responsible for the hydrolyses of α-galactosides contained in oligosaccharides to galactose and sucrose [65] and α-glucosidase and β-glucosidase are responsible for the breakdown β-glucans and cellulose [66]. In the cecum, the effect of the additives used was much smaller. Only the combined use of XYL + EMU significantly influenced the increase in β-glucosidase activity (P < 0.0001) and to a small extent on the increase in β-glucuronidase activity (P < 0.05). β-glucosidase is an enzyme that hydrolyses some NSPs and ferments starch [67, 68]. It seems that the use of xylanase, release some of the nutrients  providing substrates for fermentation by bacteria; however, the increase in β-glucuronidase activity is interesting. Studies show that the high activity of this enzyme leads to negative changes in the intestinal microflora, which is indicative of the population of E. coli and Clostridium growth,  [69, 70]. During the experiment, the mortality of birds did not differ from the average and during the sampling, no areas affected by disease in the gastrointestinal tract were noticed.

Line 441-442: Increase in activity does not mean high activity. Is the enzyme activity in the referenced paper on the same level as in this dataset or much higher? Please only speculate about infection if the activity is similar, as otherwise you do not have further evidence substantiating the idea, as you also do describe.

It is well known that this data are incomparable between studies, of course, we add a sentence that it is only speculation  and now we have:

Hübener et al. [7] presented studies in which an increase in the number of glycolytic bacteria in a similar situation was observed. It could be speculated that in the current study, the addition of XYL counteracted to some extent the negative effect of NSP on the viscosity and digestibility of carbohydrates caused a decrease in the activity of bacterial enzymes by lowering the amount of substrate for them.